# Microbes display broad diversity in cobamide preferences

Kenny C. Mok,[1] Olga M. Sokolovskaya,[1] Adam M. Deutschbauer,[1,2] Hans K. Carlson,[2] Michiko E. Taga[1]

**ABSTRACT** Cobamides, the vitamin $B_{12}$ (cobalamin) family of cofactors, are used by most organisms but produced by only a fraction of prokaryotes, and are thus considered key shared nutrients among microbes. Cobamides are structurally diverse, with multiple different cobamides found in most microbial communities. The ability to use different cobamides has been tested for several bacteria and microalgae, and nearly all show preferences for certain cobamides. This approach is limited by the commercial unavailability of cobamides other than cobalamin. Here, we have extracted and purified seven commercially unavailable cobamides to characterize bacterial cobamide preferences based on growth in specific cobamide-dependent conditions. The tested bacteria include engineered strains of *Escherichia coli*, *Sinorhizobium meliloti*, and *Bacillus subtilis* expressing native or heterologous cobamide-dependent enzymes, cultured under conditions that functionally isolate specific cobamide-dependent processes such as methionine synthesis. Comparison of these results to those of previous studies of diverse bacteria and microalgae revealed that a broad diversity of cobamide preferences exists not only across different organisms but also between different cobamide-dependent metabolic pathways within the same organism. The microbes differed in the cobamides that support growth most efficiently, cobamides that do not support growth, and the minimum cobamide concentrations required for growth. The latter differ by up to four orders of magnitude across organisms from different environments and by up to 20-fold between cobamide-dependent enzymes within the same organism. Given that cobamides are shared, required for use of specific growth substrates, and essential for central metabolism in certain organisms, cobamide preferences likely impact community structure and function.

**IMPORTANCE** Nearly all bacteria are found in microbial communities with tens to thousands of other species. Molecular interactions such as metabolic cooperation and competition are key factors underlying community assembly and structure. Cobamides, the vitamin $B_{12}$ family of enzyme cofactors, are one such class of nutrients, produced by only a minority of prokaryotes but required by most microbes. A unique aspect of cobamides is their broad diversity, with nearly 20 structural forms identified in nature. Importantly, this structural diversity impacts growth as most bacteria that have been tested show preferences for specific cobamide forms. We measured cobamide-dependent growth in several model bacteria and compared the results to those of previous analyses of cobamide preference. We found that cobamide preferences vary widely across bacteria, showing the importance of characterizing these aspects of cobamide biology to understand the impact of cobamides on microbial communities.

**KEYWORDS** cobalamin, cobamide, cobamide preference, corrinoid, vitamin $B_{12}$

$B_{12}$ and other cobamide cofactors are required by organisms in all domains of life (1). Only certain bacteria and archaea synthesize cobamides (2), while others must acquire them exogenously. Thus, cobamides are considered shared nutrients within microbial communities. A unique aspect of cobamides is their structural diversity.

Address correspondence to Michiko E. Taga, taga@berkeley.edu.

The authors declare no conflict of interest.

See the funding table on p. 6.

While B$_{12}$ (cobalamin, Cbl) is a well-known vitamin important for human health, nearly 20 cobamides with alternative lower ligands exist in nature (Fig. 1A) (3–5). Though research with alternate cobamides is limited because they are commercially unavailable, cobamide-dependent enzymes and organisms are known to have distinct preferences for different cobamides (6–16). Further, the addition of different cobamides to soil or soil-derived enrichment cultures elicited distinct shifts in bacterial abundances, suggesting that cobamide structure influences bacterial growth at the community level (17). Thus, cobamide preference in bacteria is likely important for microbial community structure.

Here, we address four questions about cobamide use and preference. (i) Do different cobamide-dependent processes in the same bacterium have the same cobamide requirements? (ii) How many, and which, cobamides can support growth? iii) Which cobamides are preferred, and do preferences differ across organisms? (iv) How much cobamide do microbes need for growth, and do these requirements vary by the cobamide-dependent pathway, taxonomy, or environment? We addressed these questions by measuring the growth of wild-type and engineered bacteria in different cobamide-dependent conditions with up to eight cobamides at a range of concentrations. Cobamide preference was defined based on the concentration of the added cobamide that elicits half-maximal growth (EC$_{50}$), with lower values corresponding to more preferred cobamides. This *in vivo* assay encompasses the different aspects of cobamide utilization, including uptake, adenosylation, riboswitch-based regulation, and the cobamide-dependent enzymes themselves. We compared these measurements with those from published literature to gain a comprehensive view of *in vivo* cobamide preferences across taxa and environments.

## Different cobamide-dependent enzymes in a single organism can have distinct cobamide preferences

First, we compared the cobamide requirements of three cobamide-dependent processes in *Sinorhizobium meliloti* engineered to lack cobamide biosynthesis capability. Wild-type *S. meliloti* produces Cbl for methionine synthase (MetH), methylmalonyl-CoA mutase (MCM), and ribonucleotide reductase (NrdJ), each of which can be tested separately for cobamide preference by altering the genetic background and growth substrates (see supplemental methods). Although MetH-dependent growth was supported by all cobamides tested, the benzimidazolyl cobamides Cbl and [Bza]Cba were strongly preferred, with EC$_{50}$ values two to three orders of magnitude lower than those of the purinyl and phenolyl cobamides (Fig. 1B; Table S1). These concentration requirements and preferences are similar to what we previously observed for MCM-dependent growth (8), except that [2-MeAde]Cba better supported MCM-dependent growth (Fig. 2). NrdJ-dependent growth required 25- and 15-fold higher Cbl and [Bza]Cba concentrations, respectively (Fig. 1B and C; Table S1), which suggests either more cobamide is required for NrdJ function or higher levels of NrdJ enzyme are necessary to support growth. The relative cobamide preferences for MetH- and NrdJ-dependent growth are similar, though phenolyl cobamides did not support NrdJ-dependent growth, consistent with NrdJ being a base-on enzyme (18) and phenolyl cobamides being unable to adopt the base-on conformation (Fig. 1A through C and 2). Cbl was the preferred cobamide for all three cobamide-dependent conditions, consistent with adaptation to endogenously produced Cbl (Fig. 2).

We used *Escherichia coli*, a bacterium unable to synthesize cobamides *de novo*, to compare cobamide preferences of ethanolamine ammonia-lyase (EAL)- and MetH-dependent growth using engineered strains cultured under different conditions. *E. coli* requires EAL for growth when ethanolamine is the nitrogen source and requires MetH for methionine synthesis when *metE*, encoding the cobamide-independent methionine synthase, is deleted. EAL-dependent growth required 5- to 78-fold higher cobamide concentrations than MetH-dependent growth and showed different cobamide preferences (Fig. 1D and E; Table S1). [2-MeAde]Cba, produced by *E. coli* when provided the

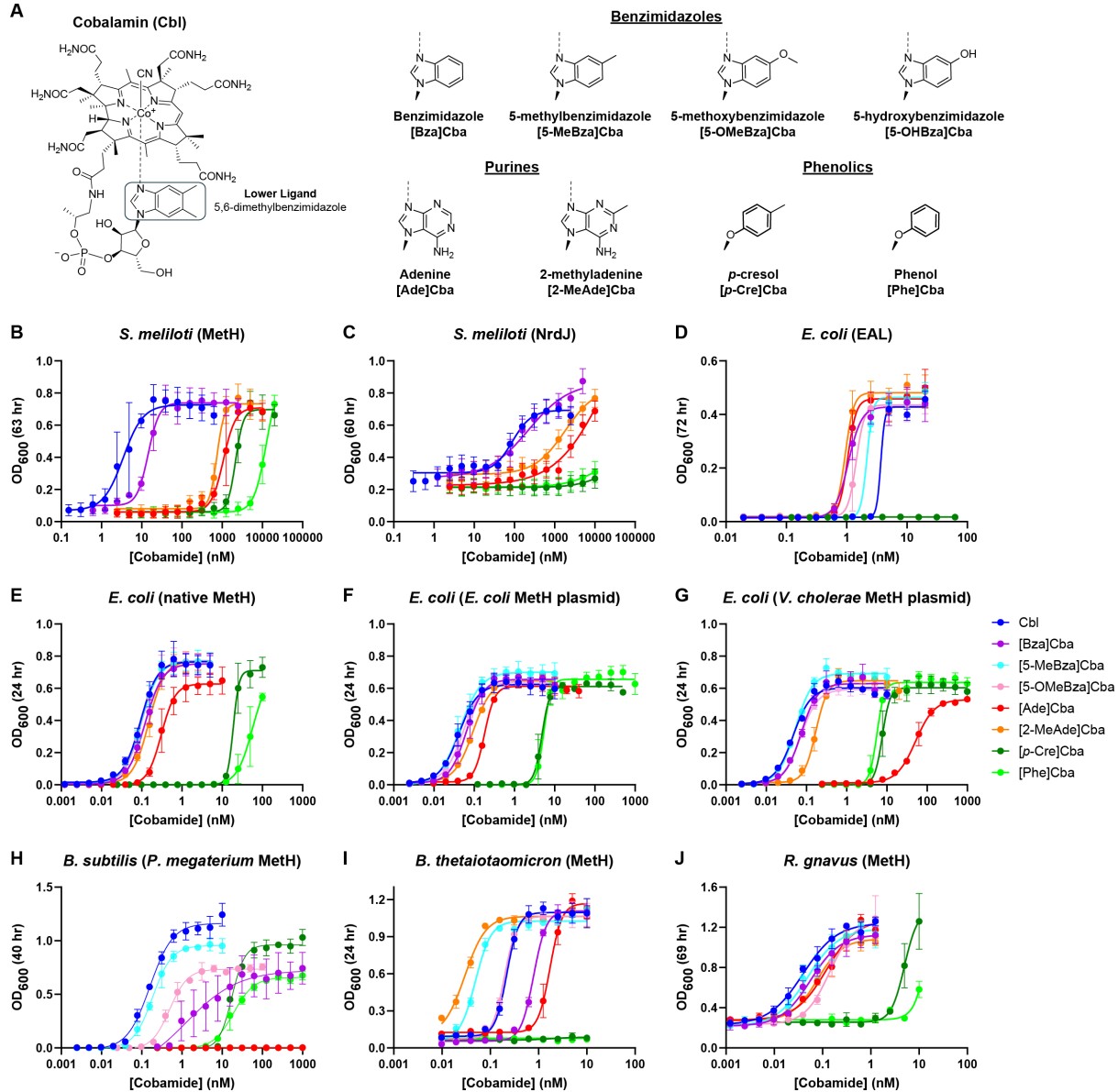

**FIG 1** Cobamide structures and cobamide-dependent growth. (A) Structure of B$_{12}$ (cobalamin), which contains the lower ligand 5,6-dimethylbenzimidazole, is shown in the base-on conformation in which the ring nitrogen is coordinated to the cobalt ion (dashed line) (left). Alternative lower ligands of cobamides in this study (right) are shown, with the names of the cobamides given below each structure. (B–J) Cobamide dose-dependent growth assays showing OD$_{600}$ measured at the indicated times for (B) MetH-dependent growth of *S. meliloti*, (C) NrdJ-dependent growth of *S. meliloti*, (D) EAL-dependent growth of *E. coli*, (E) MetH-dependent growth of *E. coli*, (F) MetH-dependent growth of *E. coli* expressing *E. coli metH* on a plasmid, (G) MetH-dependent growth of *E. coli* expressing *V. cholerae metH* on a plasmid, (H) MetH-dependent growth of *B. subtilis* expressing *P. megaterium metH*, (I) MetH-dependent growth of *B. thetaiotaomicron*, and (J) MetH-dependent growth of *R. gnavus*. The EC$_{50}$ values calculated from the curves in panels B–J and genotypes of the engineered strains in B–H are shown in Table S1. OD$_{600}$ values for [Bza]Cba-supplemented cultures of *B. subtilis* were recorded after 72 hours because growth was not observed until after 45 hours. Points represent the means of 3–6 biological replicates; error bars represent standard deviation.

precursor cobinamide (22), was most preferred for EAL-dependent growth, but less preferred for MetH-dependent growth. In contrast, Cbl was the least preferred cobamide that supports EAL-dependent growth, but most preferred for MetH-dependent growth (Fig. 1D and E). [*p*-Cre]Cba did not support EAL-dependent growth, consistent with EAL being a base-on enzyme (10). Together, these results show that cobamide-dependent pathways in the same organism can have distinct cobamide preferences and require different cobamide concentrations.

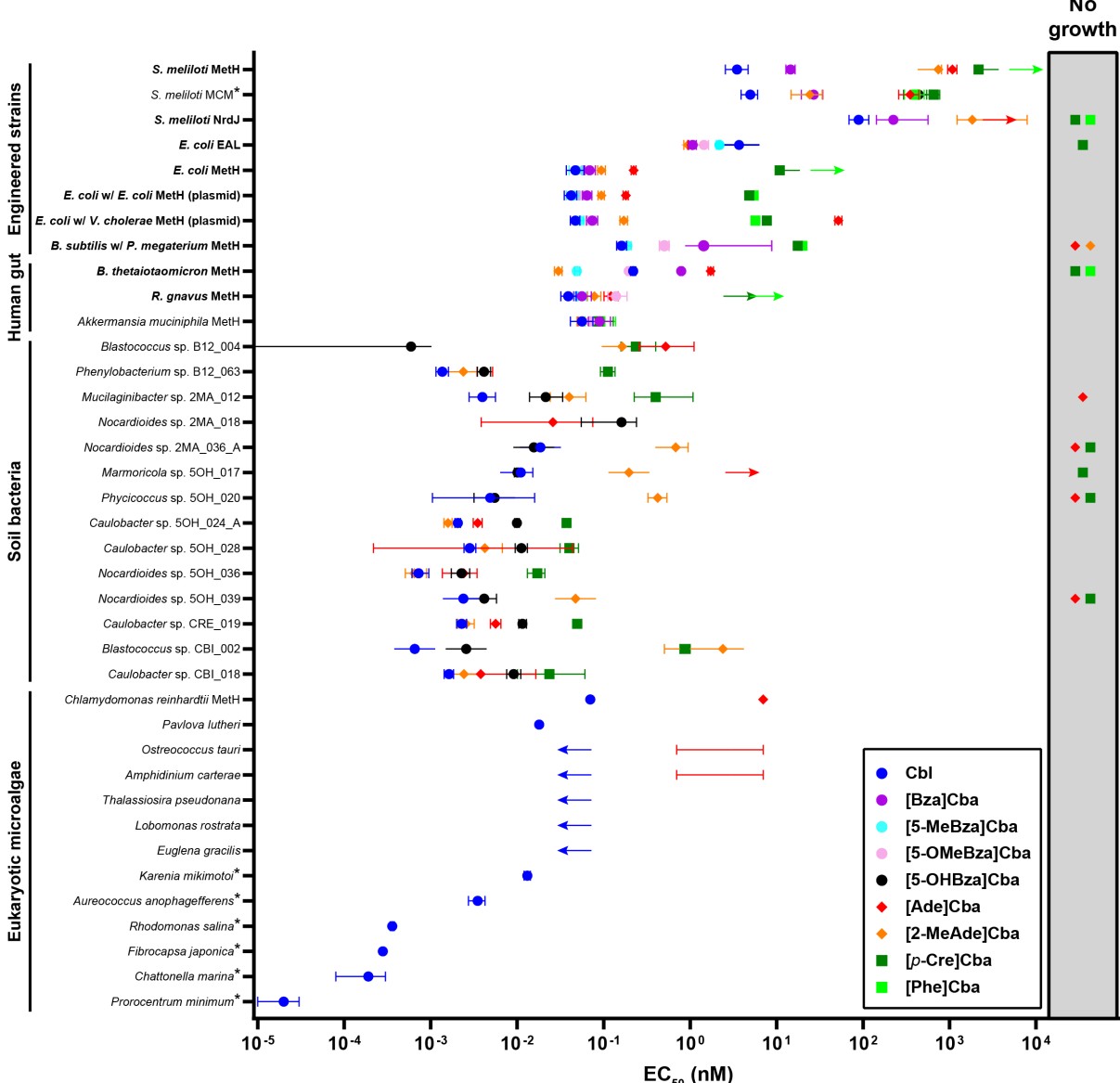

**FIG 2** Comparison of EC$_{50}$ values for cobamide-dependent growth. Organisms examined in the current study (bold) are compared with those from previous studies (8, 13, 19–21). *S. meliloti* Rm1021 Δ*nrdJ cobD::gus* Gm$^R$ *metH::*Tn5 pMSO3-*nrdAB*(*E. coli*) was used for *S. meliloti* MCM-dependent growth (8). EC$_{50}$ values for *S. meliloti* are higher than for the other tested microbes likely because wild-type *S. meliloti* synthesizes Cbl *de novo* and lacks a high-affinity cobamide uptake system. MetH-dependent growth of *C. reinhardtii* was tested in a *metE* mutant (13). Symbols show the mean EC$_{50}$ values. Capped bars represent 95% confidence intervals, except with organisms labeled with *, which indicates error as standard deviation. Errors of *C. reinhardtii* and *P. lutheri* EC$_{50}$ values were not reported (13). Bars are uncapped on the left or right when lower or upper bounds for 95% confidence intervals could not be determined, respectively. The lower bound for *Blastococcus* sp. B12_004 grown with [5-OHBza]Cba is $10^{-7}$ nM (21). The base of the leftward and rightward arrows represents maximal and minimal concentrations for EC$_{50}$ from dose–response assays in which lack of growth or saturating growth was not reached, respectively. For *O. tauri* and *A. carterae*, EC$_{50}$ values could not be calculated, but the capped bars for [Ade]Cba show the upper and lower bounds (13). Symbols in the shaded region on the right represent cobamides that were unable to support growth at any concentration tested.

## Cobamide use in a single process varies across bacteria

We next examined cobamide preferences for MetH-dependent growth in several additional bacteria. These included two cobamide non-producers that inhabit the human gut, *Bacteroides thetaiotaomicron* and *Ruminococcus gnavus*, which have MetH but lack MetE, as well as engineered *E. coli* and *Bacillus subtilis* strains heterologously expressing MetH orthologs from *Vibrio cholerae* and *Priestia megaterium*, respectively. In

each strain, growth was supported by all or most of the cobamides tested, suggesting there is promiscuity in cobamide uptake, adenosylation, regulation, and use by MetH (Fig. 1E through J). However, $EC_{50}$ values spanned a minimum of two orders of magnitude for each individual strain among cobamides supporting growth (Fig. 1 and 2). While in all cases, either [Ade]Cba or a phenolyl cobamide was least preferred, and Cbl was most preferred, except by *B. thetaiotaomicron*, variability in cobamide preference was observed across organisms (Fig. 1F through J). This variability is most apparent when contrasting *B. thetaiotaomicron* and *B. subtilis* expressing *P. megaterium metH*, which did not grow with phenolyl or purinyl cobamides, respectively, at any tested concentration (Fig. 1H and I). The results of the latter strain largely mirror prior growth measurements at a single cobamide concentration (23), and the preference for Cbl is consistent with *P. megaterium* being a Cbl producer. In contrast, we previously showed that the human gut microbe *Akkermansia muciniphila* has no cobamide preference (Fig. 2) due to its ability to remodel diverse cobamides to [Ade]Cba (19).

Expressing *V. cholerae metH* in *E. coli* afforded the opportunity to compare the cobamide preferences of *E. coli* and *V. cholerae* MetH orthologs in the same intracellular environment. Unlike *E. coli*, wild-type *V. cholerae* cannot use [Ade]Cba for MetH-dependent growth (12). Our results suggest this is due to its poor use by MetH as the $EC_{50}$ for [Ade]Cba was nearly 300-fold higher for *E. coli* expressing *V. cholerae metH* compared to its native *metH* (52 versus 0.18 nM) (Fig. 1F and G; Table S1). Overexpression of *E. coli metH* led to improved growth with the least preferred cobamides (Fig. 1E and F). This suggests MetH is limiting for *E. coli* growth with certain cobamides including [Ade]Cba and is consistent with our previous observation that mutation of the regulator *metR* or the *metH* 5′ untranslated region improved the growth of *E. coli* with [Ade]Cba (24). Overall, even with a limited number of taxa, we observed considerable variability in cobamide use with MetH-dependent growth in different bacteria. While generally, benzimidazolyl cobamides were most preferred and phenolyl cobamides the least, growth with specific cobamides varied greatly between organisms, particularly purinyl cobamides.

## Cobamide requirements in bacteria and microalgae span orders of magnitude and correspond to the environment

Comparison of these results to those of other studies that evaluated cobamide-dependent growth demonstrated that taxonomically diverse microbes from soil are also variable in their relative preferences for different cobamides and that Cbl requirements of eukaryotic microalgae vary by nearly four orders of magnitude (Fig. 2). Further, the lowest $EC_{50}$ values for most soil bacteria and microalgae are 1–2 orders of magnitude lower than those for human gut commensal bacteria (Fig. 2). This suggests the former are adapted to survive at much lower cobamide concentrations, which could be due to more efficient use of cobamides through improvements to processes such as uptake or binding by cobamide-dependent enzymes. If $EC_{50}$ values are indicative of cobamide concentrations in these environments, the bioavailable cobamide concentration can be estimated as 0.1 to 10 pM in aquatic environments, 1–10 pM in soil, and 10–100 pM in the human gut. Consistent with these values, Cbl has been detected at pM concentrations in aquatic environments (25). A concentration of 41 nM cobamide has been detected in soil, though soil's robust ability to adsorb cobamides suggests not all may be bioavailable (17). Cobamide concentrations in the gut regions that harbor the bacteria in this study have not been measured. It is worth noting that while the cobamides we tested supported growth in nearly all cases (Fig. 1), the less preferred cobamides for certain bacteria are unlikely to be present in sufficient quantities to support the growth of these bacteria in their natural environments.

Together, these results demonstrate that cobamide preferences of enzymes in the same organism and of cobamide-dependent growth across taxonomically diverse microbes are variable. Furthermore, cobamide concentrations required for growth vary by orders of magnitude across environments, suggesting microbes are adapted to

cobamide levels in their environment, likely by tuning the sensitivity of cobamide uptake, adenosylation, regulation, and use by cobamide-dependent enzymes. The characterization of cobamide abundances, bioavailability, requirements, and preferences is therefore necessary to understand the role of cobamide metabolism within microbial communities.

## ACKNOWLEDGMENTS

We thank members of the Taga lab for helpful discussions and Zoila Alvarez-Aponte and Rebecca Procknow for critical reading of the manuscript. We thank Kristen LeGault and Kim Seed for genomic DNA of *Vibrio cholerae* O1 biovar El Tor strain N16961.

This research was supported by NIH grant R35GM139633 to M.E.T. A.M.D. and H.K.C. were supported by ENIGMA-Ecosystems and Networks Integrated with Genes and Molecular Assemblies (http://enigma.lbl.gov), a Science Focus Area Program at Lawrence Berkeley National Laboratory, the research is based upon work supported by the U.S. Department of Energy, Office of Science, Office of Biological and Environmental Research, under contract number DE-AC02-05CH11231.

## AUTHOR AFFILIATIONS

[1]Department of Plant and Microbial Biology, University of California, Berkeley, Berkeley, California, USA
[2]Environmental Genomics and Systems Biology Division, Lawrence Berkeley National Laboratory, Berkeley, California, USA

## PRESENT ADDRESS

Olga M. Sokolovskaya, Department of Biological Engineering, Massachusetts Institute of Technology, Cambridge, Massachusetts, USA

## AUTHOR ORCIDs

Kenny C. Mok  http://orcid.org/0000-0002-5227-6987
Michiko E. Taga  http://orcid.org/0000-0002-9148-5925

## FUNDING

| Funder | Grant(s) | Author(s) |
| --- | --- | --- |
| National Institutes of Health | R35GM139633 | Kenny C. Mok |
| National Institutes of Health | R35GM139633 | Olga M. Sokolovskaya |
| National Institutes of Health | R35GM139633 | Michiko E. Taga |
| U.S. Department of Energy | DE-AC02-05CH11231 | Adam M. Deutschbauer |
| U.S. Department of Energy | DE-AC02-05CH11231 | Hans K. Carlson |

## AUTHOR CONTRIBUTIONS

Kenny C. Mok, Conceptualization, Data curation, Formal analysis, Investigation, Methodology, Resources, Validation, Visualization, Writing – original draft | Olga M. Sokolovskaya, Conceptualization, Data curation, Formal analysis, Investigation, Methodology, Resources, Validation, Writing – review and editing | Adam M. Deutschbauer, Funding acquisition, Project administration, Resources, Supervision, Writing – review and editing | Hans K. Carlson, Methodology, Project administration, Supervision, Writing – review and editing | Michiko E. Taga, Conceptualization, Funding acquisition, Methodology, Project administration, Resources, Supervision, Writing – original draft

## ADDITIONAL FILES

The following material is available online.

### Supplemental Material

**Supplemental text (mSystems01407-24-s0001.docx).** Supplemental methods.
**Table S1 (mSystems01407-24-s0002.docx).** $EC_{50}$ values calculated from data in Fig. 1B-J.

### Open Peer Review

**PEER REVIEW HISTORY (review-history.pdf).** An accounting of the reviewer comments and feedback.

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
