## [Reviewer comments · mSystems]

Microbes display broad diversity in cobamide preferences

Kenny Mok, Olga Sokolovskaya, Adam Deutschbauer, Hans Carlson, and Michiko Taga

Corresponding Author(s): Michiko Taga, University of California Berkeley

Review Timeline:

Submission Date:	November 1, 2024
Editorial Decision:	February 11, 2025
Revision Received:	February 21, 2025
Accepted:	February 24, 2025

Editor: Rosie Alegado

Reviewer(s): The reviewers have opted to remain anonymous.

Transaction Report:

DOI: <https://doi.org/10.1128/msystems.01407-24>

Re: mSystems01407-24 (Microbes display broad diversity in cobamide preferences)

Dear Dr. Michiko E Taga:

I apologize for the extreme delay in returning reviews to you - this was exacerbated by the holidays. I hope that this will not deter you from submitting to mSystems in the future.

Revision Guidelines

Sincerely,
Rosie Alegado
Editor
mSystems

Reviewer #2 (Comments for the Author):

This work compares the cobamide preferences for cobamide-dependent reactions within and across organisms using well-designed genetic and nutritional supplementation experiments to differentiate NrdJ, EAL, and MetH function. Furthermore, a larger comparison of cobamide requirements for organisms found in different environments is made to make broader hypotheses about cobamide availability in nature. Although the figures are detailed and rich, the core text describing them is

relatively vague. As a brief example, there seems to be a hierarchy of cobamide function in *S. meliloti*, with both MetH and NrdJ having lower EC50 values with benzimidazole containing cobamides, followed by purines and finally phenolic cobamides. In the text, however, this is glossed over by saying preferences were similar to a previous study with MCM-dependent growth. Generally, more detail in the text would make this work considerably stronger, especially with the comparison of cobamide preferences for MetH in different organisms and a longer discussion about the implications of the availability of cobamides in different environments.

Minor comments:

In the methods section, please clarify the growth conditions for *Bacillus subtilis*. 0.02% glucose seems too low for a carbon source and 0.2% histidine seems too high for just satisfying an auxotrophy.

Reviewer #3 (Comments for the Author):

Mok et al. studied the impacts of a range of cobamide variants on the growth of cobamide-dependent bacterial strains. The authors demonstrated differential cobamide preferences in different strains or different metabolic processes in the same host. The manuscript is well-written and the findings are essential for understanding the impact of cobamide bioavailability on microbial community structure and ecosystem functions.

Minor comments:

1. Line 70-71. *Sinorhizobium meliloti* is capable of de novo B12 biosynthesis. Is the test of different cobamides on *Sinorhizobium meliloti* growth so meaningful? For example, the concentrations of phenol type cobamides listed in Table S1 are way too high and unrealistic in the environment. Is *Sinorhizobium meliloti* able to uptake phenol type cobamides?

2 At least some concentration numbers should be included in the main text to facilitate the readers to understand.

3 Line 88, I doubt *E. coli* is a suitable cobamide-dependent model organism to study the effects of cobamide structural variation on cobamide-dependent growth. The cobamide uptake system in *E. coli* is not so efficient and the results could be misleading. My experience is that you don't need to add any cobamide to grow an *E. coli* culture, which could mean that a cobamide is not that so important to *E. coli*. Have the authors measured the expression of MetH and EAL? Are these proteins abundantly expressed or minorly expressed?

4 Line 99-100, are *Bacteroides thetaiotaomicron* and *Ruminococcus gnavus* corrinoid auxotrophs or prototrophs? I think the rationales to use these two strains for the testing should be explicitly explained to make the results meaningful and reliable.

5 Line 124-125, this section does not very much make sense. The authors tried to infer intracellular cobamide concentrations from environmental concentration or the cobamide provided to a culture. Many corrinoid auxotrophs possess the high-affinity but cobamide uptake system and the concentrated intracellular cobamides could be orders of magnitude higher than the environmental concentrations. Have the authors measured/estimated the intracellular cobamide concentrations?

6 Overall, the ability to uptake cobamides with different lower bases is not taken into consideration in this study. The authors should address this issue. Also I believe the corrinoid prototrophs are much less sensitive by the cobamide conditions in the real environment, which brings the rationales why testing microbes such as *Sinorhizobium meliloti*.

We thank the reviewers for carefully assessing our work. We have addressed their concerns below.

Reviewer #2 (Comments for the Author):

This work compares the cobamide preferences for cobamide-dependent reactions within and across organisms using well-designed genetic and nutritional supplementation experiments to differentiate NrdJ, EAL, and MetH function. Furthermore, a larger comparison of cobamide requirements for organisms found in different environments is made to make broader hypotheses about cobamide availability in nature. Although the figures are detailed and rich, the core text describing them is relatively vague. As a brief example, there seems to be a hierarchy of cobamide function in *S. meliloti*, with both MetH and NrdJ having lower EC50 values with benzimidazole containing cobamides, followed by purines and finally phenolic cobamides. In the text, however, this is glossed over by saying preferences were similar to a previous study with MCM-dependent growth. Generally, more detail in the text would make this work considerably stronger, especially with the comparison of cobamide preferences for MetH in different organisms and a longer discussion about the implications of the availability of cobamides in different environments.

As the editor has allowed us to expand our word count, we have been able to add more detail and context to the manuscript. For changes to the topics of concern to the reviewer, please refer to lines 75-78, 129-132, 140-142, and 148-151.

Minor comments:

In the methods section, please clarify the growth conditions for *Bacillus subtilis*. 0.02% glucose seems too low for a carbon source and 0.2% histidine seems too high for just satisfying an auxotrophy.

We thank the reviewer for noticing this error. We corrected the concentrations to 0.5% glucose and 0.03% histidine.

Reviewer #3 (Comments for the Author):

Mok et al. studied the impacts of a range of cobamide variants on the growth of cobamide-dependent bacterial strains. The authors demonstrated differential cobamide preferences in different strains or different metabolic processes in the same host. The manuscript is well-written and the findings are essential for understanding the impact of cobamide bioavailability on microbial community structure and ecosystem functions.

Minor comments:

1. Line 70-71. *Sinorhizobium meliloti* is capable of de novo B12 biosynthesis. Is the test of different cobamides on *Sinorhizobium meliloti* growth so meaningful? For example, the concentrations of phenol type cobamides listed in Table S1 are way too high and unrealistic in the environment. Is *Sinorhizobium meliloti* able to uptake phenol type cobamides?

It is true that *S. meliloti* is a B₁₂ producer and unlikely to utilize alternative cobamides that may be present in the environment, and would not encounter cobamides at the required concentrations, as the reviewer points out. However, we believe it is of interest to examine the relative cobamide preferences of this bacterium because it encodes three cobamide-dependent enzymes, each of which can be studied independently due to the genetic tractability of *S. meliloti*. A comparison of preferences of different cobamide-dependent enzymes within the same organism has not previously been performed. We have added a sentence (lines 148-150) to address the reviewer's point about environmental concentrations: "It is worth noting that while the cobamides we tested supported growth in nearly all cases (Fig. 1), the less preferred cobamides for certain bacteria are unlikely to be present in sufficient quantities to support growth of these bacteria in their natural environments."

2 At least some concentration numbers should be included in the main text to facilitate the readers to understand.

Since different bacteria and metabolisms have different cobamide concentration requirements, in most cases comparing EC₅₀s is not as informative as comparing the relative preferences for the different cobamides. However, our comparison of *E. coli* and *V. cholerae* MetHs was performed in the same host strain, so we have added the EC₅₀ values for [Ade]Cba for each in line 124.

3 Line 88, I doubt *E. coli* is a suitable cobamide-dependent model organism to study the effects of cobamide structural variation on cobamide-dependent growth. The cobamide uptake system in *E. coli* is not so efficient and the results could be misleading. My experience is that you don't need to add any cobamide to grow an *E. coli* culture, which could mean that a cobamide is not that so important to *E. coli*. Have the authors measured the expression of MetH and EAL? Are these proteins abundantly expressed or minorly expressed?

The reviewer is correct that *E. coli* generally does not require cobamides for growth. This is because while it has MetH, it also encodes for MetE, the cobamide-independent methionine synthase. Therefore, we used an engineered strain of *E. coli* in which *metE* was deleted to test MetH-dependent growth. As for EAL, many cobamide-dependent enzymes are necessary only during growth with specific substrates, in this case ethanolamine as the nitrogen source. We have added these details in lines 91-93. "*E. coli* requires EAL for growth when ethanolamine is the nitrogen source and requires MetH for methionine synthesis when *metE*, encoding the cobamide-independent methionine synthase, is deleted."

It is not clear to us why the reviewer believes the cobamide uptake system of *E. coli* is not efficient, but they may be referring to *E. coli* B strains, many of which have a defect in cobamide uptake due to a stop codon in the *btuB* gene encoding the outer membrane corrinoid transporter (doi.org/10.1016/j.jmb.2009.09.021). We are using a K strain that has a functional uptake system. Regardless, we stated in line 64 that our *in vivo* growth assay takes into account all aspects of cobamide utilization, including uptake. Even if there are differences in transport of different cobamides in *E. coli* (or other bacteria that we studied), growth differences with different cobamides indicate the organisms' overall cobamide requirements.

4 Line 99-100, are *Bacteroides thetaiotaomicron* and *Ruminococcus gnavus* corrinoid auxotrophs or prototrophs? I think the rationales to use these two strains for the testing should be explicitly explained to make the results meaningful and reliable.

We have added in line 104 that *B. thetaiotaomicron* and *R. gnavus* cannot produce cobamides *de novo*. We tested these two strains, as well as the others in this section, in order to examine the MetH-dependent growth of additional organisms, which we stated in line 103.

5 Line 124-125, this section does not very much make sense. The authors tried to infer intracellular cobamide concentrations from environmental concentration or the cobamide provided to a culture. Many corrinoid auxotrophs possess the high-affinity *btu* cobamide uptake system and the concentrated intracellular cobamides could orders of magnitude higher than the environmental concentrations. Have the authors measured/estimated the intracellular cobamide concentrations?

While we did correlate EC₅₀ values with environmental concentrations of cobamides, our intention was not to infer intracellular cobamide concentrations. Our *in vivo* growth assays established the extracellular cobamide concentrations necessary for growth of these bacteria in pure culture, which we used as a proxy for the required environmental cobamide levels for these bacteria. The EC₅₀ values are not a measure of the intracellular cobamide concentration that is necessary for growth. To emphasize this point, we have inserted “of added cobamide” into our description of EC₅₀ in lines 62-64: “Cobamide preference was defined based on the concentration of added cobamide that elicits half-maximal growth (EC₅₀), with lower values corresponding to more preferred cobamides.”

6 Overall, the ability to uptake cobamides with different lower bases is not taken into consideration in this study. The authors should address this issue. Also I believe the corrinoid prototrophs are much less sensitive by the cobamide conditions in the real environment, which brings the rationales why testing microbes such as *Sinorhizobium meliloti*.

It is true that we did not specifically discuss the effect of uptake on cobamide preference. However, as discussed in response to comment 1, we have stated in the text that EC₅₀ incorporates uptake, as well as other aspects of cobamide utilization, since we used a growth-based assay. Only in cases where the enzymes were expressed in the same host (MetH, NrdJ, and MCM in *S. meliloti*; *E. coli* and *V. cholerae* MetH's in *E. coli*) did we speculate that differences in EC₅₀ were due to differences in the enzymes themselves.

For *S. meliloti*, if the concern about sensitivity is in regard to the 100-1000X higher cobamide concentrations required for growth, *S. meliloti*, unlike the other bacteria in this study, lacks a high-affinity cobamide uptake system. This was noted in the legend of Figure 1. In addition, many other producers have cobamide-binding riboswitches that control cobamide biosynthesis gene expression, down-regulating expression in the presence of exogenous cobamides. Thus, even though they are capable of synthesizing cobamides, many cobamide producers will preferentially use cobamides present in the environment.

Re: mSystems01407-24R1 (Microbes display broad diversity in cobamide preferences)

Dear Dr. Michiko E Taga:

Your manuscript has been accepted, and I am forwarding it to the ASM production staff for publication. Your paper will first be checked to make sure all elements meet the technical requirements. ASM staff will contact you if anything needs to be revised before copyediting and production can begin. Otherwise, you will be notified when your proofs are ready to be viewed.

Sincerely,

Rosie Alegado
Editor
mSystems